# Future-Aware End-to-End Driving: Bidirectional Modeling of Trajectory Planning and Scene Evolution

**Bozhou Zhang[1,2], Nan Song[1,2], Jingyu Li[1,2], Xiatian Zhu[3]\*, Jiankang Deng[4], Li Zhang[1,2]\***

[1]School of Data Science, Fudan University     [2]Shanghai Innovation Institute
[3]University of Surrey     [4]Imperial College London

https://github.com/LogosRoboticsGroup/SeerDrive

## Abstract

End-to-end autonomous driving methods aim to directly map raw sensor inputs to future driving actions such as planned trajectories, bypassing traditional modular pipelines. While these approaches have shown promise, they often operate under a one-shot paradigm that relies heavily on the current scene context, potentially underestimating the importance of scene dynamics and their temporal evolution. This limitation restricts the model's ability to make informed and adaptive decisions in complex driving scenarios. We propose a new perspective: the future trajectory of an autonomous vehicle is closely intertwined with the evolving dynamics of its environment, and conversely, the vehicle's own future states can influence how the surrounding scene unfolds. Motivated by this bidirectional relationship, we introduce **SeerDrive**, a novel end-to-end framework that jointly models future scene evolution and trajectory planning in a closed-loop manner. Our method first predicts future bird's-eye view (BEV) representations to anticipate the dynamics of the surrounding scene, then leverages this foresight to generate future-context-aware trajectories. Two key components enable this: (1) future-aware planning, which injects predicted BEV features into the trajectory planner, and (2) iterative scene modeling and vehicle planning, which refines both future scene prediction and trajectory generation through collaborative optimization. Extensive experiments on the NAVSIM and nuScenes benchmarks show that SeerDrive significantly outperforms existing state-of-the-art methods.

## 1 Introduction

End-to-end autonomous driving [1] has emerged as a promising paradigm by jointly learning perception [2, 3, 4], prediction [5, 6, 7, 8], and planning [9, 10, 11] in a unified framework. Compared to traditional modular pipelines, this approach simplifies system design and enables planning-oriented optimization through holistic training. Recent advances [12, 13, 14, 15, 16] have demonstrated that directly generating future trajectories from raw sensor inputs can achieve strong performance, highlighting the potential of end-to-end methods for building scalable and efficient driving systems. These methods have been widely evaluated in both open-loop [17] and closed-loop [18] settings, across real-world datasets [19] and simulation platforms [20].

Despite these successes, most existing methods adopt a one-shot paradigm, in which sensor observations, typically from the current time step, are used to directly predict a trajectory several seconds into the future. In this setup, the model must rely heavily on the scene's current situation to infer the ego

---

*Li Zhang (lizhangfd@fudan.edu.cn) and Xiatian Zhu (xiatian.zhu@surrey.ac.uk) are the corresponding authors.

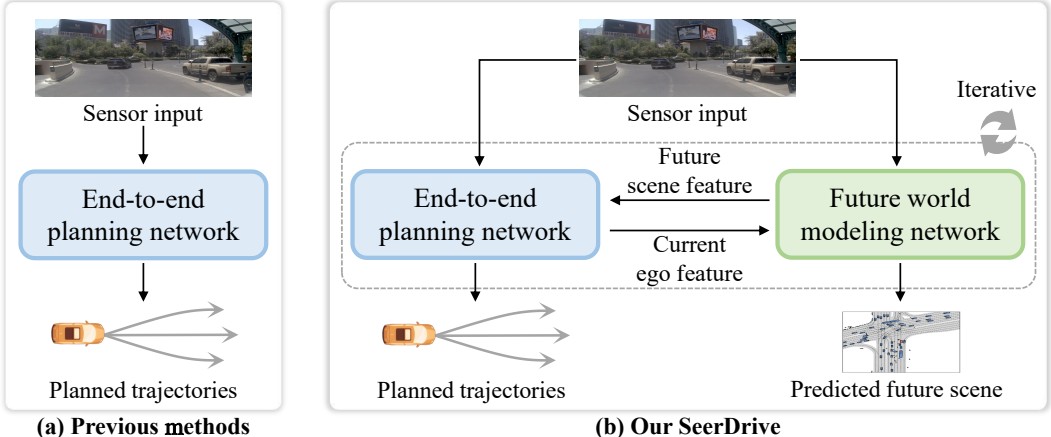

| (a) Previous methods | (b) Our SeerDrive |

Figure 1: **Paradigm comparison.** (a) Prior end-to-end methods follow a *one-shot* paradigm, directly mapping sensor inputs to planned trajectories based only on the current scene. (b) In contrast, Seer-Drive predicts scene evolution with a world model and plans trajectories with a planning model, enabling iterative interaction between the two in a closed-loop process.

vehicle's future motion (Figure 1(a)). While effective in structured environments, this approach tends to *underestimate the importance of how the scene may evolve over time*, a crucial factor in dynamic and interactive driving contexts. Furthermore, *the ego vehicle's own future actions can significantly influence how the surrounding scene unfolds*. These two aspects, the future scene and the agent's future behavior, are inherently coupled and should be modeled together. However, this bidirectional dependency remains underexplored in current end-to-end systems.

To address this gap, we draw inspiration from the emerging concept of world models, which offer the ability to learn environment dynamics and simulate future observations. We propose to leverage this capability not only to foresee how the driving scene will change, but also to coordinate it with the ego vehicle's planned actions through mutual interaction.

In this paper, we present **SeerDrive**, a novel end-to-end driving framework that introduces a paradigm shift by explicitly modeling the bidirectional relationship between scene evolution and trajectory planning (Figure 1(b)). To implement this paradigm, we design two key components. *Future-Aware Planning* injects predicted future bird's-eye view (BEV) features into the planner, enabling trajectory generation informed by both current perception and anticipated dynamics. *Iterative Scene Modeling and Vehicle Planning* refines both the predicted scene and the planned trajectory through mutual feedback, supporting adaptive and temporally consistent decision-making. Together, these components enable context-aware planning in complex, dynamic environments.

Our **contributions** are threefold. (I) We propose a new paradigm for end-to-end driving that explicitly captures the bidirectional interaction between scene dynamics and the ego vehicle's future actions, challenging the conventional one-shot planning approach. (II) We instantiate this paradigm through a unified framework, SeerDrive, which jointly models future BEV scene representations and vehicle trajectories through future-aware and iterative interaction mechanisms. (III) We conduct extensive experiments on both the NAVSIM and nuScenes benchmarks, demonstrating that SeerDrive achieves state-of-the-art performance and validates the effectiveness of our proposed design.

## 2 Related work

**End-to-end autonomous driving.** End-to-end autonomous driving [12, 13, 14, 15] has attracted increasing attention, as it contrasts with traditional modular methods by integrating perception [2, 4], prediction [5, 7], and planning [9, 11] into a unified, differentiable framework that enables end-to-end optimization. Early methods [21, 22, 23, 24] often bypass intermediate tasks and directly infer planning trajectories or actions from sensor data, both in open-loop [17] and closed-loop [20] settings. UniAD [12] unifies perception, prediction, and planning into a differentiable framework and leverages a transformer architecture to optimize the entire pipeline in a planning-oriented manner, achieving

strong performance across all tasks. VAD [13] adopts a vectorized representation to improve the efficiency of the end-to-end pipeline. VADv2 [25] introduces a probabilistic planning paradigm with a large vocabulary and demonstrates impressive performance in closed-loop settings. SparseDrive [14] leverages a sparse scene representation to make better use of temporal information. Several other studies [26, 27] explore self-supervised learning to simplify the complex end-to-end pipeline.

Recent efforts have focused increasingly on more challenging end-to-end planning benchmarks [18, 19]. DiffusionDrive [15] proposes a truncated diffusion policy to achieve more accurate and diverse planning. GoalFlow [28] incorporates flow matching and goal-point guidance into end-to-end autonomous driving. DriveTransformer [16] investigates the scaling law within a unified transformer-based architecture. Different from previous methods that focus solely on improving the planning process itself, our approach aims to jointly optimize future world modeling and planning in an adaptive manner to achieve better planning performance.

**World model in autonomous driving.** World models aim to predict future scene dynamics conditioned on the current environment and ego state. Some studies [29, 30, 31] address world modeling by training models to generate realistic videos that are consistent with physical principles. Drive-Dreamer [32] employs a latent diffusion model guided by 3D boxes, HD maps, and ego states, and introduces an additional decoder for future action prediction. Drive-WM [33] generalizes this setup to multi-camera inputs and explores its application in the end-to-end planning task. Some other methods explore improving the generalization of dynamic world modeling by scaling up both data and model architecture. GAIA-1 [34] models token sequences using an autoregressive transformer conditioned on past states, followed by a diffusion-based decoder for realistic video synthesis. Vista [35] is trained on diverse web-collected driving videos and utilizes latent diffusion to produce high-resolution, long-horizon video outputs.

In contrast to methods that focus on generating complex and realistic images, some approaches generate driving scenes in the bird's-eye view. SLEDGE [36] introduces the first generative simulator designed for agent motion planning. GUMP [37] builds upon generative modeling to capture dynamic traffic interactions, enabling diverse and realistic future scenario simulations. UMGen [38] generates ego-centric, multimodal driving scenes in an autoregressive, frame-by-frame manner. Scenario Dreamer [39] employs a vectorized latent diffusion model that directly operates on structured scene elements, coupled with an autoregressive Transformer for simulating agent behaviors in a data-driven way. Following the simplicity and structured nature of BEV, we likewise adopt it for modeling future scene evolution.

**Joint world modeling and planning.** Several works [27, 40, 41, 42, 43] explore joint modeling and collaborative optimization of world models and planning models, employing various strategies. In the field of autonomous driving, some works extend the world model by introducing an action token. OccWorld [44] employs an occupancy-based representation and jointly generates future occupancy and ego actions in an autoregressive manner. Drive-OccWorld [45] further extends this pipeline to operate directly from images, enabling 4D occupancy forecasting and planning. In contrast, some works start from the end-to-end autonomous driving pipeline and aim to enhance planning through world modeling. Existing methods either leverage world modeling as an additional supervision signal during training [26, 27], or use it to assist in selecting the best trajectory among multiple candidates [43]. Our approach conducts an in-depth exploration of world modeling and end-to-end planning design, enabling deep interaction between the two processes and significantly improving planning performance.

## 3 Methodology

### 3.1 Preliminary

**Task formulation.** End-to-end autonomous driving maps sensor inputs (e.g., camera and LiDAR) to future ego trajectories, often using multi-modal outputs to capture diverse possible futures. Auxiliary tasks like detection, map segmentation, and agent motion prediction are commonly integrated to enhance scene understanding and support safer planning. World models in autonomous driving aim to predict future scene evolution based on current observations, enabling agents to simulate and evaluate future outcomes. They offer structured representations of the environment, which help improve planning accuracy and decision reliability.

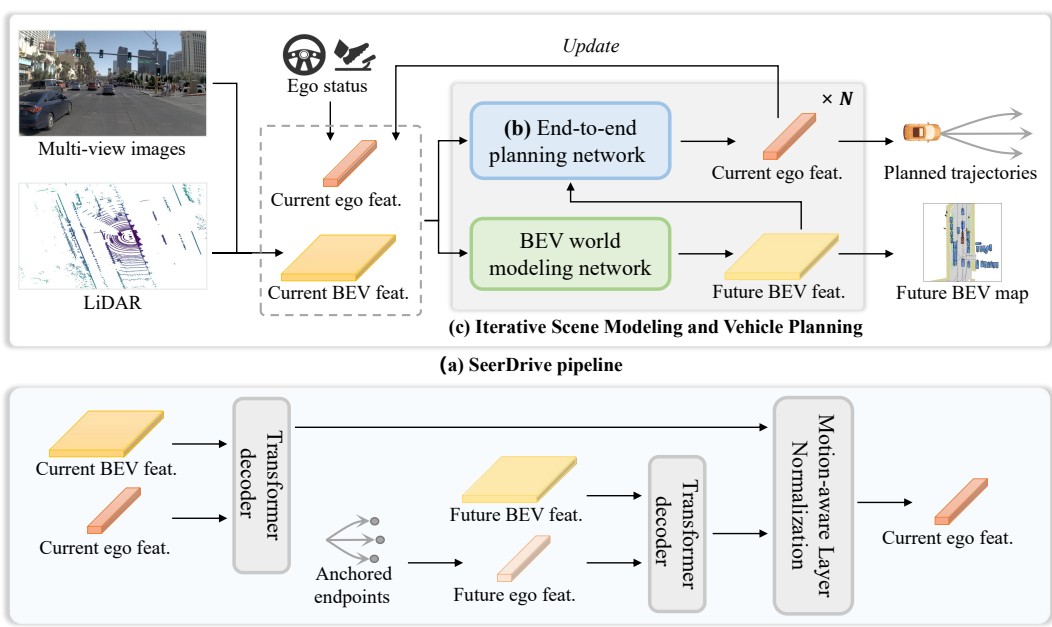

Figure 2: Overview of **SeerDrive**. (a) Multi-view images and LiDAR inputs are encoded to obtain the current BEV feature, while ego status is encoded to produce the current ego feature. These are then used to predict the future BEV feature. All three types of features are subsequently used for trajectory planning. (b) The end-to-end planning network generates future trajectories based on the current ego feature, current BEV feature, and future BEV feature. (c) The BEV world modeling network and the end-to-end planning network operate iteratively, updating the current ego feature to progressively improve planning performance.

**Framework overview.** As illustrated in Figure 2, the SeerDrive framework consists of two key components. (a) shows the overall pipeline, where multi-view images and LiDAR inputs are fused to obtain the current BEV feature, and the ego status is encoded as the current ego feature. These features are fed into the BEV world modeling and end-to-end planning networks, which operate iteratively to predict the future BEV map and generate planned trajectories. (b) details the end-to-end planning network, which incorporates the future BEV feature—produced by the world modeling network—to enhance trajectory planning. (c) depicts the iterative interaction between world modeling and planning, enabling gradual refinement and improved planning performance.

## 3.2 Feature encoding

As illustrated in Figure 2 (a), multiple types of features are encoded. Given multi-view images $\mathcal{I}$, and LiDAR features $\mathcal{P}$, the encoder transforms these multi-modal sensor inputs into a current BEV feature map $F_{\text{bev}}^{\text{curr}} \in \mathbb{R}^{H \times W \times C}$, where $H$ and $W$ are the spatial dimensions of the BEV feature, and $C$ is the number of feature channels. Following prior works [15, 43, 46], we adopt TransFuser [21] to obtain a unified BEV representation. Subsequently, a lightweight BEV decoder is employed to generate the current BEV semantic map $\mathcal{B}_{\text{curr}}$ for supervision. Following prior works [14, 15, 43], the anchored multi-modal trajectories $\mathcal{T}$ and ego status $\mathcal{E}$ are encoded with a simple multi-layer perceptron (MLP) encoder to produce the multi-modal ego feature $F_{\text{ego}}^{\text{curr}} \in \mathbb{R}^{M \times C}$, where $M$ denotes the number of trajectory modes. The process is shown below:

$$
\begin{aligned}
F_{\text{bev}}^{\text{curr}} &= \text{TransFuser}(\mathcal{I}, \mathcal{P}), \\
F_{\text{ego}}^{\text{curr}} &= \text{EgoEncoder}(\mathcal{T}, \mathcal{E}), \\
\mathcal{B}_{\text{curr}} &= \text{BEVDecoder}(F_{\text{bev}}^{\text{curr}}).
\end{aligned}
\tag{1}
$$

## 3.3 Future BEV world modeling

Given the current BEV feature and ego feature obtained above, a BEV world model is employed to predict the future BEV representation. Instead of modeling future images, which is both challenging and computationally intensive, we follow recent works [27, 36, 43] that model structured and simplified BEV representations as a more efficient alternative. The current BEV feature $F_{\text{bev}}^{\text{curr}}$ is first flattened along the spatial dimensions and then repeated across the modality dimension, resulting in an updated $F_{\text{bev}}^{\text{curr}} \in \mathbb{R}^{M \times HW \times C}$. It is then concatenated with the current ego feature $F_{\text{ego}}^{\text{curr}}$ to form the scene feature $F_{\text{scene}}^{\text{curr}} \in \mathbb{R}^{M \times (HW+1) \times C}$. The BEV world model, implemented as a Transformer encoder, produces the future scene feature $F_{\text{scene}}^{\text{fut}} \in \mathbb{R}^{M \times (HW+1) \times C}$. From this, the future BEV feature $F_{\text{bev}}^{\text{fut}}$ is extracted. A lightweight BEV decoder is then applied to generate the future BEV semantic map $\mathcal{B}_{\text{fut}}$ for supervision. The overall process is illustrated below:

$$F_{\text{scene}}^{\text{fut}} = \text{BEVWorldModel}(F_{\text{scene}}^{\text{curr}}),$$
$$\mathcal{B}_{\text{fut}} = \text{BEVDecoder}(F_{\text{bev}}^{\text{fut}}). \tag{2}$$

## 3.4 Future-aware end-to-end planning

After obtaining the current BEV feature, current ego feature, and future BEV feature, the end-to-end planning network jointly reasons over the present scene and its future evolution to generate the planned trajectories. However, enabling the planning network to simultaneously consider both the current and future BEV features is non-trivial. Directly interacting the ego feature with both can lead to entangled representations, causing confusion in ego planning and degrading performance. To address this, we adopt a decoupled strategy, where the ego feature interacts with the current and future BEV features independently, and the results are subsequently fused for final trajectory planning.

As illustrated in Figure 2 (b), the current ego feature $F_{\text{ego}}^{\text{curr}}$ interacts with the current BEV feature $F_{\text{bev}}^{\text{curr}}$ through a Transformer decoder. The updated ego feature is then passed through an MLP decoder to generate the planned trajectories $\mathcal{T}_{\text{a}}$ for supervision. Since the future BEV feature represents the future scene and corresponds to the future ego state, we initialize the future ego feature $F_{\text{ego}}^{\text{fut}}$ using the endpoints of the anchored trajectories. It then interacts with the future BEV feature $F_{\text{bev}}^{\text{fut}}$ through a Transformer decoder. Similarly, an MLP decoder is applied to produce the planned trajectories $\mathcal{T}_{\text{b}}$ for supervision. The process is shown below:

$$F_{\text{ego}}^{\text{curr}} = \text{TransformerDecoder}(F_{\text{ego}}^{\text{curr}}, F_{\text{bev}}^{\text{curr}}),$$
$$F_{\text{ego}}^{\text{fut}} = \text{TransformerDecoder}(F_{\text{ego}}^{\text{fut}}, F_{\text{bev}}^{\text{fut}}),$$
$$\mathcal{T}_{\text{a}} = \text{EgoDecoder}(F_{\text{ego}}^{\text{curr}}),$$
$$\mathcal{T}_{\text{b}} = \text{EgoDecoder}(F_{\text{ego}}^{\text{fut}}). \tag{3}$$

To incorporate the future ego feature into the current ego feature, we adopt motion-aware layer normalization (MLN) [4] to obtain a future-aware ego representation. This representation is then passed through an MLP decoder to generate the final planned trajectories $\mathcal{T}_{\text{final}}$:

$$F_{\text{ego}}^{\text{curr}} = \text{MLN}(F_{\text{ego}}^{\text{curr}}, F_{\text{ego}}^{\text{fut}}),$$
$$\mathcal{T}_{\text{final}} = \text{EgoDecoder}(F_{\text{ego}}^{\text{curr}}). \tag{4}$$

## 3.5 Iterative scene modeling and vehicle planning

As illustrated in Figure 2 (c), the BEV world modeling network and the end-to-end planning network operate in an iterative manner to progressively improve planning performance. This is motivated by the mutual dependency between future scene evolution and ego trajectories—future traffic dynamics influence the ego's motion plans, while the ego's planned actions, in turn, shape the future scene. In the end-to-end planning network, the future BEV feature $F_{\text{bev}}^{\text{fut}}$ serves as a reference for generating the planned trajectories. Meanwhile, the refined ego feature $F_{\text{ego}}^{\text{curr}}$ is fed back into the BEV world modeling network to produce an updated future BEV feature. This iterative process is repeated $N$ times, yielding $N$ pairs of predicted future semantic maps and ego trajectories, denoted as $(\mathcal{B}_{\text{fut}}^{(1)}, \mathcal{T}_{\text{a}}^{(1)}, \mathcal{T}_{\text{b}}^{(1)}, \mathcal{T}_{\text{final}}^{(1)}), \ldots, (\mathcal{B}_{\text{fut}}^{(N)}, \mathcal{T}_{\text{a}}^{(N)}, \mathcal{T}_{\text{b}}^{(N)}, \mathcal{T}_{\text{final}}^{(N)})$, which are all supervised during training.

### 3.6 End-to-end learning

The model is trained in an end-to-end manner with a loss function comprising two components: the BEV semantic map loss $\mathcal{L}_{\text{map}}$ and the planned trajectory loss $\mathcal{L}_{\text{traj}}$. The semantic map loss includes the current BEV map loss $\mathcal{L}_{\text{map}}^{\text{curr}}$ and the future BEV map losses from $N$ iterations, denoted as $\mathcal{L}_{\text{map}}^{\text{fut (1)}}, \ldots, \mathcal{L}_{\text{map}}^{\text{fut (N)}}$. The trajectory loss consists of $N$ sets of trajectory planning losses, each containing three terms as described in the end-to-end planning network: $(\mathcal{L}_{\text{traj}}^{\text{a (1)}}, \mathcal{L}_{\text{traj}}^{\text{b (1)}}, \mathcal{L}_{\text{traj}}^{\text{final (1)}}), \ldots, (\mathcal{L}_{\text{traj}}^{\text{a (N)}}, \mathcal{L}_{\text{traj}}^{\text{b (N)}}, \mathcal{L}_{\text{traj}}^{\text{final (N)}})$. The overall loss function for end-to-end training is as follows:

$$
\begin{aligned}
\mathcal{L}_{\text{map}} &= \lambda_1 \mathcal{L}_{\text{map}}^{\text{curr}} + \lambda_2 (\mathcal{L}_{\text{map}}^{\text{fut (1)}} + \cdots + \mathcal{L}_{\text{map}}^{\text{fut (N)}}), \\
\mathcal{L}_{\text{traj}} &= \lambda_3 ((\mathcal{L}_{\text{traj}}^{\text{a (1)}} + \mathcal{L}_{\text{traj}}^{\text{b (1)}} + \mathcal{L}_{\text{traj}}^{\text{final (1)}}) + \cdots + (\mathcal{L}_{\text{traj}}^{\text{a (N)}} + \mathcal{L}_{\text{traj}}^{\text{b (N)}} + \mathcal{L}_{\text{traj}}^{\text{final (N)}})), \quad (5) \\
\mathcal{L}_{\text{total}} &= \mathcal{L}_{\text{map}} + \mathcal{L}_{\text{traj}}.
\end{aligned}
$$

where $\lambda_1$, $\lambda_2$, and $\lambda_3$ are the balancing factors.

## 4 Experiments

### 4.1 Datasets and metrics

**Datasets.** We conduct experiments on two large-scale real-world autonomous driving datasets: NAVSIM [19] and nuScenes [17]. NAVSIM, built upon nuPlan [47], is designed for non-reactive simulation in complex scenarios with dynamic intention changes. It contains 1,192 training/validation (navtrain) and 136 testing (navtest) scenarios, with 8-camera and LiDAR data at 2 Hz. nuScenes includes 1,000 scenes with 6-camera and LiDAR data at 2 Hz. We follow the standard 700/150 train/validation split and evaluate planning in an open-loop setting.

**Evaluation metrics.** For the NAVSIM dataset, we use the PDM Score (PDMS), which comprises multiple sub-metrics: No At-Fault Collisions (NC), Drivable Area Compliance (DAC), Time-to-Collision (TTC), Comfort (Comf.), and Ego Progress (EP). For the nuScenes dataset, we follow the VAD [13] setting and report the L2 Displacement Error and Collision Rate.

### 4.2 Implementation details

**Model settings.** The model for the NAVSIM dataset uses both images and LiDAR as input, whereas the nuScenes model relies solely on images. For backbone networks, ResNet34 is employed for NAVSIM and ResNet50 for nuScenes. Regarding the number of camera views, 3 views are used in NAVSIM and 6 in nuScenes. The input image resolution is 1024×256 following TransFuser for NAVSIM, and 640×360 for nuScenes. The number of trajectory modes is set to 256 for NAVSIM and 6 for nuScenes. For NAVSIM, the planning horizon is 4 seconds with 8 future steps, and for nuScenes, it is 3 seconds with 6 steps. In both settings, we predict the future BEV semantic map corresponding to the final planning step.

**Training settings.** The model is trained on 8 NVIDIA GeForce RTX 3090 GPUs. For NAVSIM, the batch size is 16 per GPU, with 30 training epochs and a total training time of around 5 hours. For nuScenes, the batch size is 1 per GPU, with 12 epochs and a training time of about 12 hours. The learning rate is set to $2 \times 10^{-4}$ for NAVSIM and $1 \times 10^{-4}$ for nuScenes, both optimized using AdamW [48]. The loss balancing factors are set to $\lambda_1 = 10$, $\lambda_2 = 0.1$, and $\lambda_3 = 1$ for NAVSIM, and all set to 1 for nuScenes.

### 4.3 Comparison with state of the art

As shown in Table 1, we compare SeerDrive with several state-of-the-art methods on the NAVSIM dataset. Our approach achieves the highest PDM Score of 88.9. Under the same ResNet34 [49] backbone, SeerDrive outperforms recent methods, including the trajectory refinement model Hydra-NeXt [50], the online trajectory evaluation method WoTE [43], and the truncated diffusion policy method DiffusionDrive [15], demonstrating the effectiveness of our iterative world modeling and

Table 1: Performance comparison of planning on the NAVSIM *navtest* split with closed-loop metrics. "C & L" represents the use of both camera and LiDAR as sensor inputs. ResNet34 [49] is employed as the backbone for BEV feature extraction, following TransFuser [21]. The best and second best results are highlighted in **bold** and underline, respectively. † denotes the use of V2-99 [51] as the image backbone.

| Method | Input | NC ↑ | DAC ↑ | TTC ↑ | Comf. ↑ | EP ↑ | PDMS ↑ |
|---|---|---|---|---|---|---|---|
| VADv2-$\mathcal{V}_{8192}$ [25] | C & L | 97.2 | 89.1 | 91.6 | **100** | 76.0 | 80.9 |
| Hydra-MDP-$\mathcal{V}_{8192}$ [46] | C & L | 97.9 | 91.7 | 92.9 | **100** | 77.6 | 83.0 |
| UniAD [12] | Camera | 97.8 | 91.9 | 92.9 | **100** | 78.8 | 83.4 |
| LTF [21] | Camera | 97.4 | 92.8 | 92.4 | **100** | 79.0 | 83.8 |
| PARA-Drive [55] | Camera | 97.9 | 92.4 | 93.0 | 99.8 | 79.3 | 84.0 |
| TransFuser [21] | C & L | 97.7 | 92.8 | 92.8 | **100** | 79.2 | 84.0 |
| LAW [26] | Camera | 96.4 | 95.4 | 88.7 | 99.9 | 81.7 | 84.6 |
| DRAMA [56] | C & L | 98.0 | 93.1 | 94.8 | **100** | 80.1 | 85.5 |
| Hydra-MDP++ [57] | Camera | 97.6 | 96.0 | 93.1 | **100** | 80.4 | 86.6 |
| DiffusionDrive [15] | C & L | 98.2 | 96.2 | 94.7 | **100** | 82.2 | 88.1 |
| WoTE [43] | C & L | **98.5** | 96.8 | **94.9** | 99.9 | 81.9 | 88.3 |
| Hydra-NeXt [50] | C & L | 98.1 | **97.7** | 94.6 | **100** | 81.8 | 88.6 |
| SeerDrive | C & L | 98.4 | 97.0 | **94.9** | 99.9 | **83.2** | **88.9** |
| GoalFlow† [28] | C & L | 98.4 | 98.3 | 94.6 | 100 | 85.0 | 90.3 |
| SeerDrive† | C & L | 98.8 | 98.6 | 95.8 | 100 | 84.2 | 90.7 |

Table 2: Performance comparison of planning on the nuScenes *validation* split. ResNet50 [49] is used as the backbone for all methods, except for UniAD [12], which adopts ResNet101. "w/o bev" indicates without future BEV injection, and "w/o iter" indicates without iterative world modeling and planning.

| Method | L2 (m) ↓ | | | | Col. Rate (%) ↓ | | | |
|---|---|---|---|---|---|---|---|---|
| | 1s | 2s | 3s | Avg. | 1s | 2s | 3s | Avg. |
| ST-P3 [22] | 1.33 | 2.11 | 2.90 | 2.11 | 0.23 | 0.62 | 1.27 | 0.71 |
| BEV-Planner [54] | 0.28 | 0.42 | **0.68** | 0.46 | 0.04 | 0.37 | 1.07 | 0.49 |
| VAD-Tiny [13] | 0.46 | 0.76 | 1.12 | 0.78 | 0.21 | 0.35 | 0.58 | 0.38 |
| LAW [26] | 0.26 | 0.57 | 1.01 | 0.61 | 0.14 | 0.21 | 0.54 | 0.30 |
| PARA-Drive [55] | 0.25 | 0.46 | 0.74 | 0.48 | 0.14 | 0.23 | 0.39 | 0.25 |
| VAD-Base [13] | 0.41 | 0.70 | 1.05 | 0.72 | 0.07 | 0.17 | 0.41 | 0.22 |
| GenAD [58] | 0.28 | 0.49 | 0.78 | 0.52 | 0.08 | 0.14 | 0.34 | 0.19 |
| UniAD [12] | 0.44 | 0.67 | 0.96 | 0.69 | 0.04 | 0.08 | 0.23 | 0.12 |
| BridgeAD [53] | 0.29 | 0.57 | 0.92 | 0.59 | 0.01 | **0.05** | 0.22 | 0.09 |
| MomAD [52] | 0.31 | 0.57 | 0.91 | 0.60 | 0.01 | **0.05** | 0.22 | 0.09 |
| SparseDrive [14] | 0.29 | 0.58 | 0.96 | 0.61 | 0.01 | **0.05** | 0.18 | 0.08 |
| SeerDrive $_{w/o\ bev}$ | 0.22 | 0.49 | 0.73 | 0.48 | 0.02 | **0.05** | 0.16 | 0.08 |
| SeerDrive $_{w/o\ iter}$ | 0.28 | 0.47 | 0.81 | 0.52 | 0.02 | 0.07 | 0.20 | 0.10 |
| SeerDrive | **0.20** | **0.39** | 0.69 | **0.43** | **0.00** | **0.05** | 0.14 | **0.06** |

planning strategy. Furthermore, when replacing the ResNet-34 backbone with V2-99 [51], as done in GoalFlow [28], our method achieves better performance with significantly lower computational cost while supporting end-to-end training.

As shown in Table 2, we further evaluate our method on the nuScenes dataset. It achieves notable performance gains compared with recent state-of-the-art methods, including SparseDrive [14], MomAD [52], and BridgeAD [53]. Although the nuScenes dataset contains relatively simple scenarios and imperfect evaluation metrics [54], our method still achieves clear improvements, demonstrating the effectiveness of the two main designs.

## 4.4  Ablation study

**Effects of components.**    As shown in Table 3, we conduct an ablation study on the key components of our method, including Future-Aware Planning and Iterative Scene Modeling and Vehicle Planning. In the first row, both modules are removed, so the planned trajectories rely only on the current BEV feature and no iterative process is applied. This leads to a drop in PDMS from 88.9 to 87.1. In the second row, we remove the future BEV injection for planning, which also results in decreased performance, showing the importance of future BEV features for planning. In the third row, we remove the iterative process, and performance drops as well, indicating its role in refining the results. The last row presents the full SeerDrive model with both modules included, achieving the highest performance.

Table 3: Ablation study on the key components of our model. "Iter. S&V" refers to Iterative Scene Modeling and Vehicle Planning.

| Future-aware plan | Iter. S&V | NC ↑ | DAC ↑ | TTC ↑ | Comf. ↑ | EP ↑ | PDMS ↑ |
|---|---|---|---|---|---|---|---|
| | | 98.3 | 95.6 | 94.5 | **100** | 81.1 | 87.1 |
| | ✓ | 98.2 | 96.6 | 94.3 | **100** | 82.0 | 87.9 |
| ✓ | | 98.2 | 96.5 | 94.4 | **100** | 82.5 | 88.1 |
| ✓ | ✓ | **98.4** | **97.0** | **94.9** | 99.9 | **83.2** | **88.9** |

**Effects of Future-Aware Planning.**    As shown in Table 4, we conduct an ablation study on the design of the future-aware end-to-end planning. The first two rows analyze different strategies for incorporating the future BEV feature. In the first row, future BEV injection is removed entirely. In the second row, the decoupled strategy is omitted, and the network learns jointly from current and future BEV features. Both settings lead to notable performance degradation, highlighting the effectiveness of our proposed design. In the third and fourth rows, we analyze how the current and future ego features are combined. In the third row, the Motion-aware Layer Normalization (MLN) is removed, and the two features are concatenated and passed through an MLP for dimension alignment. In the fourth row, MLN is also removed, and the features are directly added. Both variations lead to a notable drop in performance, demonstrating the effectiveness of MLN in producing a future-aware ego feature.

Table 4: Ablation study on the design of future-aware end-to-end planning.

| Type | NC ↑ | DAC ↑ | TTC ↑ | Comf. ↑ | EP ↑ | PDMS ↑ |
|---|---|---|---|---|---|---|
| w/o Future BEV | 98.2 | 96.6 | 94.3 | **100** | 82.0 | 87.9 |
| w/o Decouple | 98.0 | 96.0 | 94.0 | 99.5 | 81.6 | 87.3 |
| MLN2Cat | 98.3 | 96.6 | 94.7 | **100** | 82.4 | 88.3 |
| MLN2Add | 98.2 | **97.0** | 94.1 | **100** | 82.9 | 88.5 |
| SeerDrive | **98.4** | **97.0** | **94.9** | 99.9 | **83.2** | **88.9** |

**Effects of the number of iterations.**    As shown in Table 5, a moderate number of iterations for scene modeling and vehicle planning achieves a good balance between efficiency and performance. We observe that using two iterations yields the best trade-off.

Table 5: Ablation study on the number of iterations for scene modeling and vehicle planning.

| Number | NC ↑ | DAC ↑ | TTC ↑ | Comf. ↑ | EP ↑ | PDMS ↑ |
|---|---|---|---|---|---|---|
| 1 | **98.4** | 96.6 | 94.5 | **100** | 82.2 | 88.1 |
| 2 | **98.4** | 97.0 | **94.9** | 99.9 | **83.2** | **88.9** |
| 3 | **98.4** | 97.2 | **94.9** | **100** | 82.5 | 88.7 |

**Prediction steps of future BEV.** We predict the future BEV only at the final planning step, as the last frame provides the most informative reference for determining the trajectory direction. This design yields an effective and efficient planning reference. As shown in Table 6, we further analyze this choice by predicting a sequence of intermediate BEV representations and concatenating them as planner inputs. However, incorporating intermediate predictions brings no significant performance gains while increasing model complexity. Therefore, predicting only the final future BEV at the last planning step proves to be the more effective and efficient strategy.

Table 6: Ablation study on the number of prediction steps for future BEV.

| Prediction steps | NC ↑ | DAC ↑ | TTC ↑ | Comf. ↑ | EP ↑ | PDMS ↑ |
|---|---|---|---|---|---|---|
| 1s-2s-3s-4s | 98.3 | 97.0 | 94.6 | **100** | 83.0 | 88.8 |
| 2s-4s | **98.4** | **97.2** | 94.8 | **100** | 83.1 | **88.9** |
| 4s (Origin) | **98.4** | 97.0 | **94.9** | 99.9 | **83.2** | **88.9** |

**The performance of $\mathcal{T}_a$, $\mathcal{T}_b$, and $\mathcal{T}_{\text{final}}$.** As shown in Table 7, we conduct an ablation study to separately evaluate the performance of $\mathcal{T}_a$, $\mathcal{T}_b$, and $\mathcal{T}_{\text{final}}$. The results show that $\mathcal{T}_{\text{final}}$ significantly outperforms both $\mathcal{T}_a$ and $\mathcal{T}_b$, indicating that each contributes meaningfully to the superior performance of the final trajectory.

Table 7: Ablation study on the performance of $\mathcal{T}_a$, $\mathcal{T}_b$, and $\mathcal{T}_{\text{final}}$.

| | NC ↑ | DAC ↑ | TTC ↑ | Comf. ↑ | EP ↑ | PDMS ↑ |
|---|---|---|---|---|---|---|
| $\mathcal{T}_a$ | 98.3 | 96.6 | 94.4 | **100** | 82.7 | 88.3 |
| $\mathcal{T}_b$ | 98.2 | 96.4 | 94.2 | 99.9 | 83.0 | 88.2 |
| $\mathcal{T}_{\text{final}}$ | **98.4** | **97.0** | **94.9** | 99.9 | **83.2** | **88.9** |

**The use of the anchored endpoints to initialize the future ego feature.** The future BEV corresponds to the final planning step. Thus, we initialize the future ego feature using the anchored endpoints, allowing it to encode prior knowledge of the ego vehicle's future state at the final planning step. For more extensive evaluation, we analyze different initialization strategies for the future ego feature. As shown in Table 8, using anchored endpoints as a prior yields the best performance.

Table 8: Ablation study on the use of the anchored endpoints to initialize the future ego feature.

| | NC ↑ | DAC ↑ | TTC ↑ | Comf. ↑ | EP ↑ | PDMS ↑ |
|---|---|---|---|---|---|---|
| Random | 98.3 | 96.9 | 94.5 | **100** | 82.8 | 88.6 |
| Anchored trajectories | 98.3 | 96.8 | 94.0 | **100** | **83.7** | 88.7 |
| Anchored endpoints (Origin) | **98.4** | **97.0** | **94.9** | 99.9 | 83.2 | **88.9** |

### 4.5 Qualitative results

As shown in Figure 3, we visualize two cases: a right turn and a left turn. In the bottom middle figure, the final planned trajectory generated by our model closely aligns with the ground-truth trajectory. The bottom right figure presents the planned multi-modal trajectories, capturing multiple possible future motions. The bottom left figure shows the predicted future BEV semantic maps, which reflect how the scene evolves and how the ego vehicle's position changes after the turning behaviors.

## 5 Conclusion

This paper presents SeerDrive, a unified end-to-end framework that combines future scene modeling and trajectory planning. Unlike traditional one-shot approaches that rely only on the current scene, SeerDrive predicts future BEV representations to guide planning. It introduces two core components: Future-Aware Planning, which incorporates predicted future context into trajectory

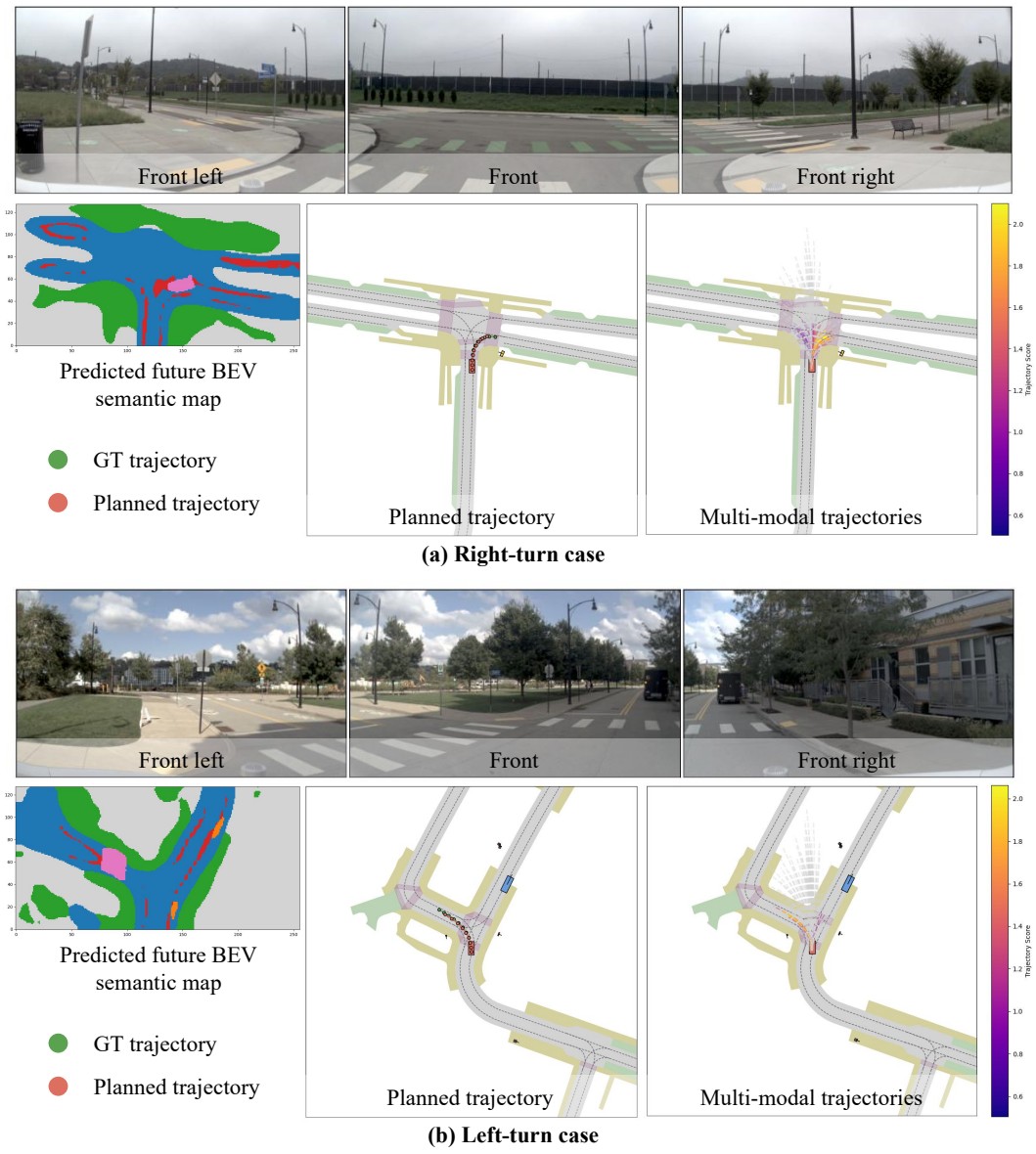

Figure 3: Qualitative results on the NAVSIM dataset. The visualization includes three front-facing camera views: front left, front, and front right, as well as model outputs including the predicted future BEV semantic map, the planned trajectory, and multi-modal predicted trajectories.

generation, and Iterative Scene Modeling and Vehicle Planning, which refines both scene predictions and plans through repeated interaction. This design enables more informed and adaptive decisions. Extensive experiments on NAVSIM and nuScenes show our approach achieves state-of-the-art results.

**Limitations and future work.**    The BEV world model adopts a transformer architecture specifically tailored to our framework, making it both effective and efficient for planning. However, it does not benefit from the generalization capabilities of foundation models. On the other hand, using off-the-shelf foundation models as world models often suffers from slow inference speed and challenges in joint optimization with the planner. Therefore, developing a tightly integrated paradigm of planning and world modeling represents a promising direction for future work.

## Acknowledgments

This work was supported in part by National Natural Science Foundation of China (Grant No. 62376060).

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

# Appendix

## A  Notations

As shown in Table 9, we provide a lookup table for notations used in the paper.

Table 9: Notations used in the paper.

| Notation | Description |
|---|---|
| $\mathcal{I}$ | multi-view images |
| $\mathcal{P}$ | LiDAR |
| $\mathcal{T}$ | anchored multi-modal trajectories |
| $\mathcal{E}$ | ego status |
| $H\&W$ | spatial dimensions of the BEV feature |
| $M$ | the number of trajectory modes |
| $C$ | the number of feature channels |
| $N$ | the number of iterations |
| $F_{\text{ego}}^{\text{curr}}$ | current ego feature |
| $F_{\text{bev}}^{\text{curr}}$ | current BEV feature map |
| $F_{\text{scene}}^{\text{curr}}$ | current scene feature |
| $F_{\text{ego}}^{\text{fut}}$ | future ego feature |
| $F_{\text{bev}}^{\text{fut}}$ | future BEV feature map |
| $F_{\text{scene}}^{\text{fut}}$ | future scene feature |
| $\mathcal{B}_{\text{curr}}$ | predicted current BEV semantic map |
| $\mathcal{B}_{\text{fut}}$ | predicted future BEV semantic map |
| $\mathcal{T}_{\text{a}}$ | planned multi-modal trajectories by current BEV |
| $\mathcal{T}_{\text{b}}$ | planned multi-modal trajectories by future BEV |
| $\mathcal{T}_{\text{final}}$ | final planned multi-modal trajectories |

## B  Implementation details

In addition to the implementation details provided in the main paper, we offer further clarifications here. The feature channels $C$ are set to 256 for both the NAVSIM [19] and nuScenes [17] datasets. Regarding the spatial dimensions $H \times W$ of the BEV features, they are $8 \times 8$ for NAVSIM and $100 \times 100$ for nuScenes. However, for future BEV map prediction, the features are uniformly downsampled to $8 \times 8$ in nuScenes. For the anchored multi-modal trajectories, we follow previous works [14, 15, 43] and use the K-Means algorithm to cluster them from the ground-truth trajectories.

## C  Evaluation metrics

In the NAVSIM dataset, trajectories spanning 4 seconds at 2Hz are first produced and then upsampled to 10Hz using an LQR controller. These refined trajectories are evaluated using a set of closed-loop metrics, which include No At-Fault Collisions ($S_{\text{NC}}$), Drivable Area Compliance ($S_{\text{DAC}}$), Time to Collision with bounds ($S_{\text{TTC}}$), Ego Progress ($S_{\text{EP}}$), Comfort ($S_{\text{CF}}$), and Driving Direction Compliance ($S_{\text{DDC}}$). The overall performance score is computed by aggregating these individual metrics. However, due to implementation limitations, $S_{\text{DDC}}$ is excluded from the final evaluation[1].

$$S_{\text{PDM}} = S_{\text{NC}} \times S_{\text{DAC}} \times s_{\text{TTC}} \times$$
$$\left( \frac{5 \times S_{\text{EP}} + 5 \times S_{\text{CF}} + 2 \times S_{\text{DDC}}}{12} \right). \tag{6}$$

---

[1]https://github.com/autonomousvision/navsim/issues/14

# D  Qualitative results

In addition to the qualitative results, we provide additional visualizations in Figure 4 and Figure 5 to further demonstrate the effectiveness of our model.

# E  Failure cases

Although our SeerDrive demonstrates strong performance, it still encounters some failure cases.

As illustrated in Figure 6 (a), the model fails to choose the correct lane after a right turn. In the right subfigure of (a), which shows the multi-modal planning result, one of the predicted trajectories closely aligns with the ground truth. However, the classification score fails to select it, suggesting that the multi-modal trajectory selection process requires further improvement.

As illustrated in Figure 6 (b), the model fails to infer the correct driving intention for a right turn. Most of the planned trajectories instead tend to change lanes to the left and proceed straight. This suggests that incorporating high-level driving intentions, such as explicit driving commands, is necessary for achieving more accurate planning outcomes.

# F  Discussions

**Comparison with existing methods.**  Table 10 summarizes the differences between our work and prior approaches in terms of how predicted future scenes from world models are utilized.

Table 10: Comparison between SeerDrive and existing methods.

| Method | Usage of predicted future scene from world model |
| --- | --- |
| LAW [26] | Used as an auxiliary supervision signal during training |
| SSR [27] | Used as an auxiliary supervision signal during training |
| WoTE [43] | Used to assist in selecting the best trajectory among multiple candidates |
| SeerDrive (Ours) | Used as feature-level reference for planning, enabling iterative interaction with planner |

**Experiment on complex scenarios.**  Following the reviewer's valuable suggestions, we evaluate our model on the test split of NAVSIM [19] under scenarios with more than 10 agents. As shown in Table 11, our model still demonstrates strong performance.

Table 11: Experiment on complex scenarios.

| | NC ↑ | DAC ↑ | TTC ↑ | Comf. ↑ | EP ↑ | PDMS ↑ |
| --- | --- | --- | --- | --- | --- | --- |
| Complex scenarios | 98.2 | 96.8 | 94.3 | 99.9 | 83.1 | 88.5 |

**Experiment beyond NAVSIM and nuScenes.**  Following the reviewer's valuable suggestions, we have further evaluated the more complex Bench2Drive [18] dataset with many rare and high-entropy scenarios such as emergency braking, merging, and overtaking, along with a longer planning horizon and closed-loop testing. As shown in Table 12, our model consistently achieves the best performance.

**Uncertainty in future predictions and planning.**  To tackle the uncertainty caused by the multi-modal characteristics of future trajectories, our framework performs prediction and planning while explicitly considering the multi-modal nature of the future. The predicted future BEV feature is multi-modal and shares the same modality as the ego feature used for planning. Accordingly, we also predict a multi-modal future BEV representation. For supervision, we adopt a winner-takes-all strategy: the best trajectory and the corresponding future BEV are selected using the same probability generated from the current ego feature, and the loss is computed based on the selected mode.

**Parameter size and inference cost.**  We evaluate our model on the NAVSIM [19] dataset. The model has 66 M parameters, and the average inference time is 24 ms under the same configuration as in the main paper.

Table 12: Experiment on the Bench2Drive [18] dataset.The first metric corresponds to open-loop testing, while the latter two are used for closed-loop evaluation.

| Method | Avg. L2 Error ↓ | Driving Score ↑ | Success Rate (%) ↑ |
|---|---|---|---|
| VAD [13] | 0.91 | 42.35 | 15.00 |
| UniAD-Tiny [12] | 0.80 | 40.73 | 13.18 |
| UniAD-Base [12] | 0.73 | 45.81 | 16.46 |
| MomAD [52] | 0.82 | 47.91 | 18.11 |
| BridgeAD [53] | 0.71 | 50.06 | 22.73 |
| SeerDrive (Ours) | **0.66** | **58.32** | **30.17** |

**Quantitative metric for predicted future BEV.** The mIoU of the predicted future BEV map on the NAVSIM [19] dataset is 38.87.

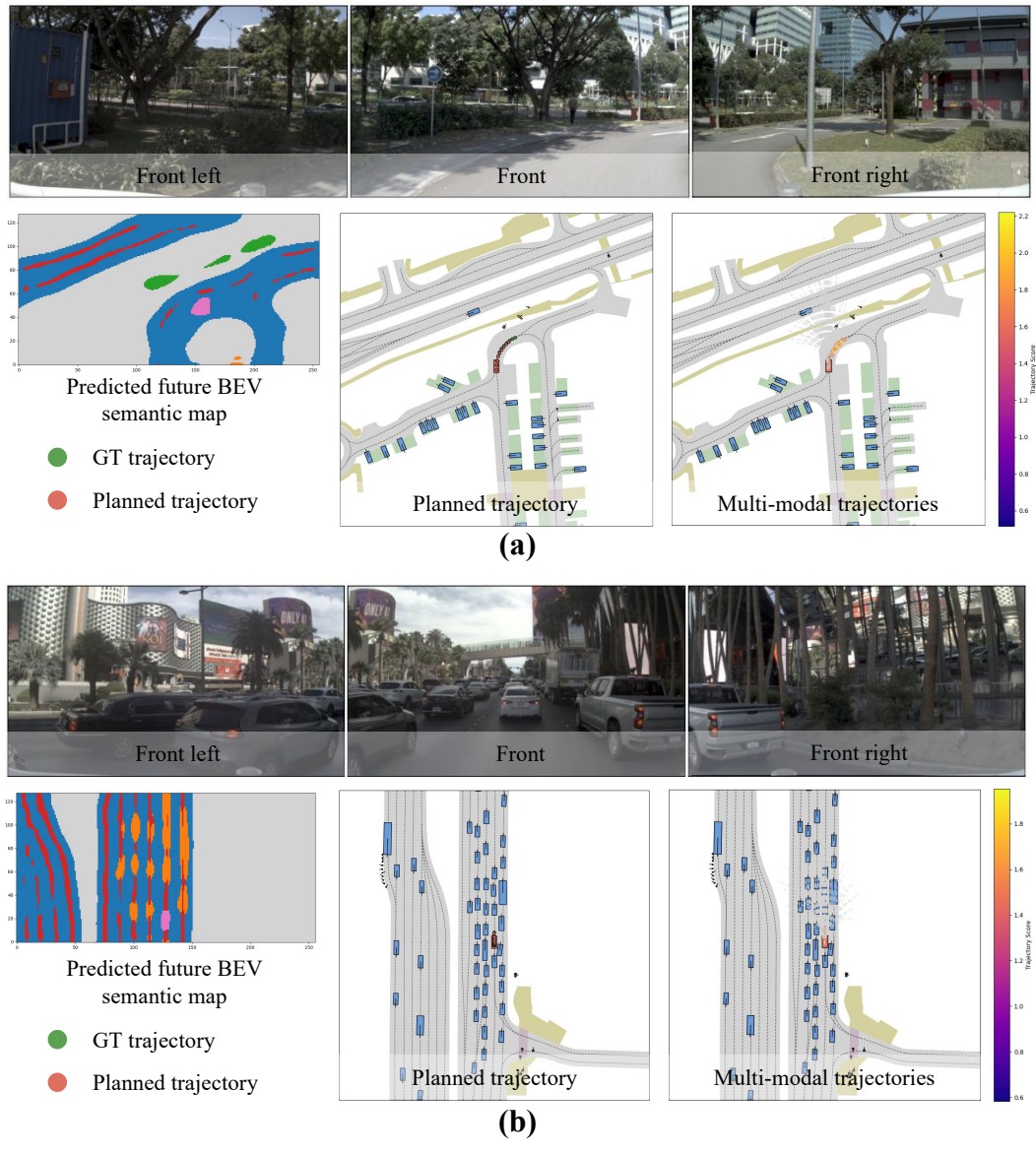

Figure 4: Qualitative results 1 on the NAVSIM dataset. The visualization includes three front-facing camera views: front left, front, and front right, as well as model outputs including the predicted future BEV semantic map, the planned trajectory, and multi-modal predicted trajectories.

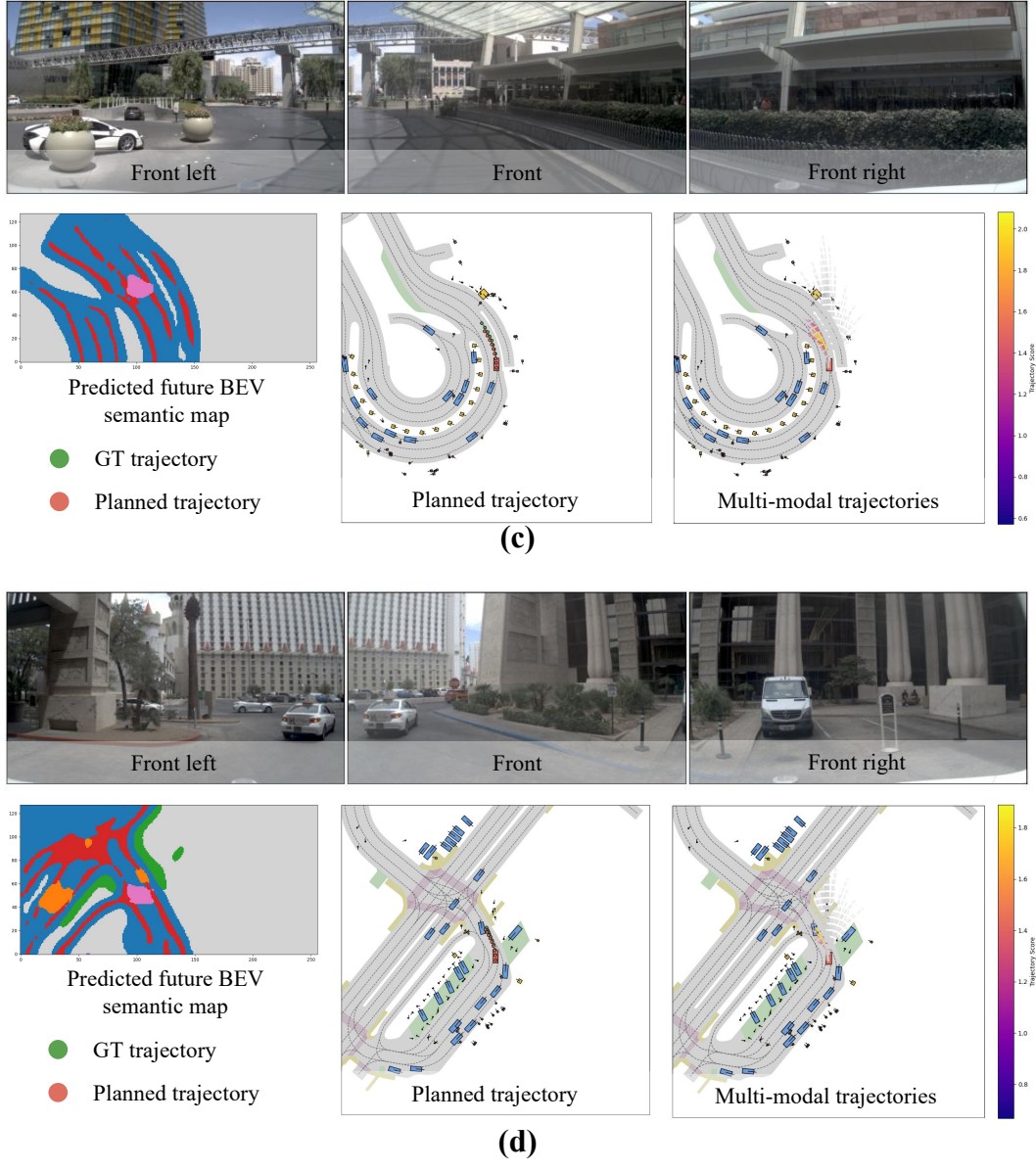

Figure 5: Qualitative results 2 on the NAVSIM dataset. The visualization includes three front-facing camera views: front left, front, and front right, as well as model outputs including the predicted future BEV semantic map, the planned trajectory, and multi-modal predicted trajectories.

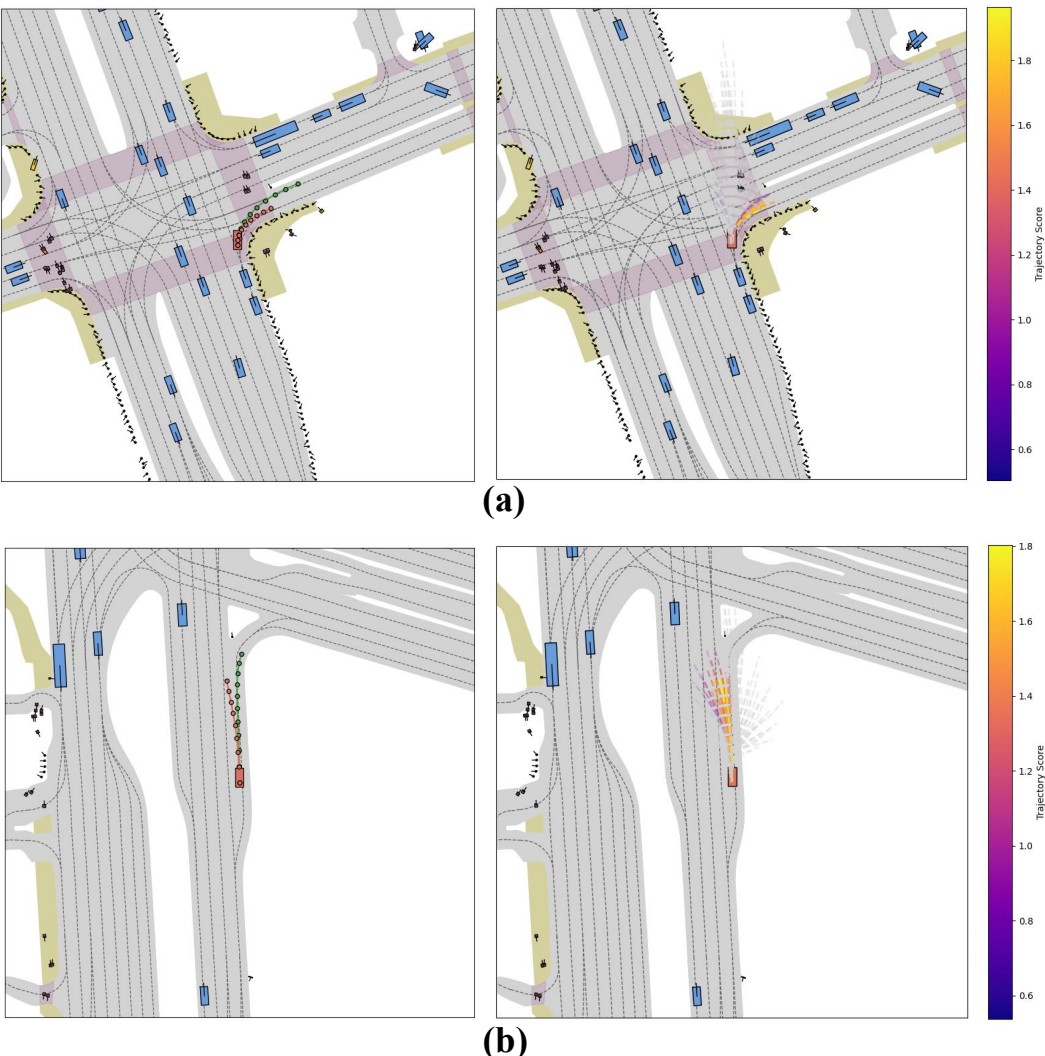

Figure 6: Failure cases from the NAVSIM dataset. In the left figure, the ground-truth trajectory is shown in green, while the top-ranked planning trajectory, selected based on the classification score, is displayed in orange. The right figure illustrates the predicted multi-modal trajectories.

