# OpenReview forum: "Future-Aware End-to-End Driving: Bidirectional Modeling of Trajectory Planning and Scene Evolution"
_NeurIPS.cc/2025/Conference — NeurIPS 2025 poster_

### Official Review · Reviewer_bKKZ · 2025-06-11

**Clarity:** 3
**Significance:** 2
**Originality:** 2
**Rating:** 4
**Confidence:** 3

**Summary:**

This paper presents SeerDrive, an end-to-end driving framework that integrates trajectory planning and future scene prediction via bidirectional modelling. Unlike traditional one-shot models that plan based solely on the current state, SeerDrive predicts future Bird's-Eye View (BEV) semantic features and injects them into the planning process, creating a feedback loop between scene evolution and ego-motion. Experiments on NAVSIM and nuScenes demonstrate significant gains over state-of-the-art methods in planning accuracy.

**Questions:**

* Can SeerDrive handle rare or high-entropy scenarios (for example crowded intersections)? Have the authors tested the model on edge cases beyond NAVSIM and nuScenes?

* Could we quantify the predictive accuracy of the future BEV semantic maps directly? This would help validate that the BEV predictions are not just beneficial but also semantically plausible.

* What is the wall-clock inference time per iteration? Could this approach be deployed real-time?

**Ethical Concerns:**

["NO or VERY MINOR ethics concerns only"]

**Final Justification:**

After carefully reading the other reviews and the rebuttal, I've decided to increase my rating to borderline accept.

**Limitations:**

yes

**Paper Formatting Concerns:**

/

**Quality:**

2

**Strengths And Weaknesses:**

### __Strengths__

1. The paper demonstrates consistent improvements over competitive baselines on NAVSIM and nuScenes, with robust closed-loop metrics.

2. The use of BEV maps rather than raw pixels for future prediction, offering a more efficient and interpretable modelling space.

3. Good ablations on the future aware planning module, and the iterative refinement.

### __Weaknesses__
1. While used as an intermediate representation, the paper does not thoroughly evaluate or visualise the quality of the future BEV predictions in isolation (e.g. with intersection-over-union).

2. The results are presented on NAVSIM and nuScenes, but it remains unclear how well SeerDrive generalises to longer horizons or more complex urban settings (e.g. large occlusions, heavy traffic).

3. The model does not clearly demonstrate that more iterations lead to better representations or insights, especially since only 2 iterations are optimal, and going to 3 harms performance.

---

> ### Author Rebuttal · Authors · 2025-07-30
>
> We thank the reviewer for the detailed review and the suggestions for improvement. Below are our responses to the reviewer’s comments:
>
> ### **Weakness 1 and Question 2: Evaluation and visualization of future BEV prediction quality.**
> Great suggestion! For visualization, we have already presented the predicted future BEV maps in Figure 3 of main paper, as well as in Figures 1 and 2 of appendix. Regarding the quality, we have now measured the mIoU of the predicted future BEV map on the NAVSIM dataset, standing at 38.87. We will include this result in the revision.
>
> ### **Weakness 2 and Question 1: Testing on rare or high-entropy scenarios, longer horizons, complex urban environments, and edge cases beyond NAVSIM and nuScenes.**
>
> - **Experiment on complex scenarios**: Thanks. To conduct the suggested test, we test our model using the scenarios from the test split of NAVSIM with more than 10 agents. As shown in the table below, our model can still achieve strong performance. We have provided a visualization in Figure 1(b) of appendix and will add more in the revision.
>
> |   | NC ↑ | DAC ↑ | TTC ↑ | EP ↑ | PDMS ↑ |
> | :--- | :---: | :---: | :---: | :---: | :---: |
> | Complex scenarios | 98.2 | 96.8 | 94.3 | 83.1 | 88.5 |
>
> - **Experiment beyond NAVSIM and nuScenes**: For more extensive evaluation, we have further evaluated the more complex Bench2Drive [1] dataset with many rare and high-entropy scenarios such as emergency braking, merging, and overtaking, along with a longer planning horizon and closed-loop testing. As shown in the table below, our model consistently achieves the best performance. We will add this test.
>
> | Method                                   | Open-loop Avg. L2 Error ↓ | Closed-loop Driving Score ↑ | Closed-loop Success Rate (%) ↑ |
> |------------------------------------------|:----------------------------:|:------------------------------:|:----------------------------------:|
> | VAD [7]                  | 0.91                       | 42.35                        | 15.00                            |
> | UniAD-Tiny [8]              | 0.80                       | 40.73                        | 13.18                            |
> | UniAD-Base [8]              | 0.73                       | 45.81                        | 16.46                            |
> | MomAD [9]              | 0.82                       | 47.91                        | 18.11                            |
> | BridgeAD [10]              | 0.71                       | 50.06                        | 22.73                            |
> | **SeerDrive (Ours)**                     | **0.66**                   | **58.32**                    | **30.17**                        |
>
> ### **Weakness 3: Iteration number and its impact on performance.**
>
> - **Explanation**: Our model design is, in fact, generic and not tied to a specific optimal iteration count. The ideal number of iterations largely depends on the characteristics and complexity of the target scenario. While more iterations generally lead to better representations, an excessive number can introduce computational burden and uncertainty. This doesn't conflict with our design; instead, our iterative architecture offers the flexibility to improve results by adding more iterations as scene complexity demands.
>
> - **Experiment**:  For instance, 2 iterations proved sufficient for NAVSIM given its comparatively lower complexity. However, for the more intricate Bench2Drive [6] dataset, our ablation study (see table below) clearly shows that 3 iterations yield superior results. This demonstrates the adaptability of our approach.
>
> | Iteration                                   | Open-loop Avg. L2 Error ↓ | Closed-loop Driving Score ↑ | Closed-loop Success Rate (%) ↑ |
> |:------------------------------------------:|:----------------------------:|:------------------------------:|:----------------------------------:|
> | 2                 | 0.74                       | 55.96                        | 27.27                            |
> | 3                 | 0.66                   | 58.32                    | 30.17                        |
> | 4                 | 0.68                       | 58.63                        | 29.98                            |
>
> Ultimately, our framework allows for a flexible trade-off between accuracy and efficiency, letting users decide the optimal balance based on their specific needs and the complexity of the environment they're working with. We will discuss this in the revision.
>
> ### **Question 3: Inference time and suitability for real-time deployment.**
>
> Thanks and we will discuss this in revision:
> Our model achieves an average inference time of 24 ms on the NAVSIM dataset, comparable to DiffusionDrive [1] — a lightweight and deployable diffusion-based model. Meanwhile, our model outperforms DiffusionDrive [1] in terms of planning performance, indicating its suitability for real-time deployment with superior effectiveness.
>
> >[1] DiffusionDrive: Truncated Diffusion Model for End-to-End Autonomous Driving, CVPR 2025.
>
> >[6] Bench2Drive: Towards Multi-Ability Benchmarking of Closed-Loop End-To-End Autonomous Driving, NeurIPS 2024.
>
> >[7] VAD: Vectorized Scene Representation for Efficient Autonomous Driving, ICCV 2023.
>
> >[8] Planning-oriented Autonomous Driving, CVPR 2023.
>
> >[9] Don't Shake the Wheel: Momentum-Aware Planning in End-to-End Autonomous Driving, CVPR 2025.
>
> >[10] Bridging Past and Future: End-to-End Autonomous Driving with Historical Prediction and Planning, CVPR 2025.

---

> > ### Comment · Reviewer_bKKZ · 2025-08-05
> >
> > Thank you very much for your responses to my questions regarding quantitative BEV prediction, higher entropy scenarios and inference time.
> >
> > After carefully reading the other reviews and the rebuttal, I've decided to increase my rating to borderline accept.

---

### Official Review · Reviewer_TxmL · 2025-07-01

**Clarity:** 2
**Significance:** 3
**Originality:** 2
**Rating:** 4
**Confidence:** 4

**Summary:**

This paper proposes SeerDrive, an end-to-end autonomous driving framework that explicitly models the bidirectional relationship between future scene evolution and ego trajectory planning. The method first predicts the future bird’s-eye-view (BEV) representation of the environment using a transformer-based world model, and then incorporates this predicted future into the planning module. Moreover, the authors propose an iterative refinement strategy where the planning and world modeling modules are jointly optimized in a closed-loop manner. Experiments on NAVSIM and nuScenes show that SeerDrive achieves state-of-the-art performance, and ablation studies demonstrate the effectiveness of its two core components: future-aware planning and iterative scene modeling.

**Questions:**

1. Is $\mathcal{T}_{\text{final}}$ a single trajectory or a set of trajectories? If it represents multiple candidates, how is the final trajectory selected from multiple candidates?

2. Please provide an ablation study that separately evaluates the performance of $\mathcal{T}_a$, $\mathcal{T}_{b}$, and $\mathcal{T}_{\text{final}}$ as described in Section 3.5. This would help clarify the contribution of each trajectory output in the overall planning process.

3. In the implementation, why is the future ego feature initialized using the anchored endpoints? Given that the endpoints are fixed while the future ego feature is updated iteratively, this design choice seems inconsistent and warrants further clarification.

**Ethical Concerns:**

["NO or VERY MINOR ethics concerns only"]

**Final Justification:**

The authors’ rebuttal has addressed my concerns. I therefore maintain my positive assessment.

**Limitations:**

yes

**Quality:**

3

**Strengths And Weaknesses:**

Strengths:
1. The paper proposes a bidirectional modeling scheme between scene evolution and trajectory planning, challenging the prevalent one-shot planning paradigm. This is a fresh and timely perspective in end-to-end driving research.

2. The framework is well-structured, with clear component design including BEV feature modeling, future-aware planning via decoupled interaction, and iterative refinement.

3. SeerDrive shows competitive or superior performance on both NAVSIM and nuScenes benchmarks. The ablation studies are thorough and help validate each design component.

Weakness
1. The method bears resemblance to the existing WoTE [1] framework. The introduction and related work sections should more clearly emphasize the differences and connections between this work and WoTE [1].


[1]:End-to-end driving with online trajectory evaluation via BEV world model.

---

> ### Author Rebuttal · Authors · 2025-07-30
>
> We thank the reviewer for the detailed review and the suggestions for improvement. Below are our responses to the reviewer’s comments:
>
> ### **Weakness 1: Differences and connections with WoTE.**
>
> - As discussed in Line 104 of the main paper, WoTE [2] employs a BEV world model to generate future BEV, which are then used to select the best trajectory from a set of candidates proposed by the planner. In contrast, our SeerDrive framework establishes feature-level interaction between BEV world modeling and end-to-end planning, enabling uniquely iterative and closed-loop communication between the two components.
>
> - While both WoTE and SeerDrive leverage a BEV world model to support the planning process, their designs are fundamentally different and orthogonal. We will clarify these distinctions in the revised paper.
>
> ### **Question 1: About $\mathcal{T}\_{\text{final}}$.**
>
> - Apologies for the lack of clarity. $\mathcal{T}\_{\text{final}}$ refers to a set of candidate trajectories. A probability score is assigned to each trajectory based on the current ego feature, and the final trajectory is selected according to the highest score, consistent with prior work [1].
> - As described in Line 200 of the main paper, we also specify the number of trajectory modes. This clarification will be highlighted in the revised version of the paper.
>
> ### **Question 2: Ablation study on the performance of $\mathcal{T}\_{\text{a}}$, $\mathcal{T}\_{\text{b}}$, and $\mathcal{T}\_{\text{final}}$.**
>
> Thanks. We conduct an ablation study to separately evaluate the performance of $\mathcal{T}\_{\text{a}}$, $\mathcal{T}\_{\text{b}}$, and $\mathcal{T}\_{\text{final}}$, as shown in the table below. The results show that $\mathcal{T}\_{\text{final}}$ significantly outperforms both $\mathcal{T}\_{\text{a}}$ and $\mathcal{T}\_{\text{b}}$, indicating that each contributes meaningfully to the superior performance of the final trajectory.
>
> |  | NC ↑ | DAC ↑ | TTC ↑ | EP ↑ | PDMS ↑ |
> | :--- | :---: | :---: | :---: | :---: | :---: |
> | $\mathcal{T}\_{\text{a}}$ | 98.3 | 96.6 | 94.4 | 82.7 | 88.3 |
> | $\mathcal{T}\_{\text{b}}$ | 98.2 | 96.4 | 94.2 | 83.0 | 88.2 |
> | $\mathcal{T}\_{\text{final}}$ | 98.4 | 97.0 | 94.9 | 83.2 | 88.9 |
>
> ### **Question 3: The use of the anchored endpoints to initialize the future ego feature.**
>
> - **Explanation**: As described in Line 202 of main paper, the future BEV corresponds to the final planning step. Thus, we initialize the future ego feature using the anchored endpoints, allowing it to encode prior knowledge of the ego vehicle’s future state at the final planning step.
>
> - **Clarification**: Note that the future ego feature is not updated iteratively — only the current ego feature undergoes iterative refinement.
>
> - **Experiment**: For more extensive evaluation, we have now analyzed different initialization strategies for the future ego feature. As shown below, using anchored endpoints as a prior yields the best performance. We will add this in revision.
>
> | Initialization strategy | NC ↑ | DAC ↑ | TTC ↑ | EP ↑ | PDMS ↑ |
> | :--- | :---: | :---: | :---: | :---: | :---: |
> | Random | 98.3 | 96.9 | 94.5 | 82.8 | 88.6 |
> | Anchored trajectories | 98.3 | 96.8 | 94.0 | 83.7 | 88.7 |
> | Anchored endpoints (Origin) | 98.4 | 97.0 | 94.9 | 83.2 | 88.9 |
>
>
> >[1] DiffusionDrive: Truncated Diffusion Model for End-to-End Autonomous Driving, CVPR 2025.
>
> >[2] End-to-end driving with online trajectory evaluation via bev world model, ICCV 2025.

---

> > ### Comment · Reviewer_TxmL · 2025-08-07
> >
> > Thank you for the authors' response. I have no further concerns.

---

### Official Review · Reviewer_LXsM · 2025-07-02

**Clarity:** 3
**Significance:** 3
**Originality:** 2
**Rating:** 5
**Confidence:** 4

**Summary:**

The paper proposes SeerDrive, an end-to-end autonomous driving framework that integrates future scene understanding into trajectory planning. The key idea is to couple a bird’s-eye view (BEV) world model that predicts semantic features of the future scene with a trajectory planner, enabling the system to make informed driving decisions based on anticipated changes in the environment.

Unlike prior work that either plans based on the current scene or uses separate modules for future prediction and planning, SeerDrive introduces a future-aware planning mechanism: it injects predicted future scene features directly into the planner and iteratively refines both the world model and the planner states over multiple steps before outputting the final trajectory.

The model is trained end-to-end using a joint loss over future BEV prediction and trajectory planning. Experiments on two datasets—NAVSIM (a closed-loop simulator) and nuScenes (open-loop real-world data)—show that SeerDrive achieves state-of-the-art driving performance, particularly in terms of safety and success rate under long-horizon planning. Ablation studies demonstrate the benefits of future feature injection and iterative refinement.

**Questions:**

- How strong is the predicted future BEV? Provide mIoU of predicted lane & actor masks at t+H to quantify BEV fidelity.
- How does SeerDrive handle uncertainty in future predictions and planning?
- How sensitive is SeerDrive to the choice of loss weights (λ₁–λ₃)?

**Ethical Concerns:**

["NO or VERY MINOR ethics concerns only"]

**Final Justification:**

The response effectively addresses several of my concerns, especially about uncertainty in future predictions and planning. The approach is interesting.

**Limitations:**

yes

**Quality:**

3

**Strengths And Weaknesses:**

Strengths:
- The paper presents a well-engineered framework that effectively combines future scene prediction and trajectory planning in a closed-loop, end-to-end differentiable system.
- The experimental evaluation is thorough on both the NAVSIM and nuScenes datasets. It shows state-of-the-art results in closed-loop driving metrics.
- The proposed iterative refinement loop between the planner and world model is implemented cleanly and shown to improve performance over non-iterative baselines.
- Ablation studies address several design choices such as feature injection and iteration count.
- SeerDrive serves as a bridge between scene-centric world models and goal-directed planning, which are often treated separately.

Limitations:
- The world model only predicts the final future BEV at horizon H, not a sequence of intermediate steps, which may limit temporal expressiveness and robustness in dynamic scenes.
- No uncertainty modeling is present in the predicted BEV or planning outputs, which can be critical in real-world driving.
- The loss balancing between multiple objectives is hand-tuned, with no sensitivity analysis shown; this raises questions about stability and generalizability.
- Gains over prior work like DiffusionDrive or WoTE are relatively small (e.g., +0.3–0.8 PDMS) and may not justify the additional complexity of iterative inference and dual-task training.

---

> ### Author Rebuttal · Authors · 2025-07-30
>
> We thank the reviewer for the detailed review and the suggestions for improvement. Below are our responses to the reviewer’s comments:
>
> ### **Limitation 1: The world model predicts the final future BEV at horizon H.**
>
> - **Explanation**: We predict the final future BEV at the horizon $H$, corresponding to the last planning step, as the future scene at the final frame serves as the primary reference with the most information for determining the final direction of the trajectory. This provides an effective and efficient reference for planning.
>
> - **Experiment**:  Regardless, as suggested, we have now analyzed this design choice by additionally predicting a sequence of intermediate BEV representations and concatenating them as references for the planner. As shown in the table below, incorporating intermediate predictions does not lead to significant performance improvements, while it increases model complexity. Thus, directly predicting the final future BEV at the last planning step is a more effective and efficient choice.
>
> | Prediction steps | NC ↑ | DAC ↑ | TTC ↑ | EP ↑ | PDMS ↑ |
> | :--- | :---: | :---: | :---: | :---: | :---: |
> | 1s-2s-3s-4s | 98.3 | 97.0 | 94.6 | 83.0 | 88.8 |
> | 2s-4s | 98.4 | 97.2 | 94.8 | 83.1 | 88.9 |
> | 4s (Origin) | 98.4 | 97.0 | 94.9 | 83.2 | 88.9 |
>
> ### **Limitation 2 and Question 2: Uncertainty in future predictions and planning.**
>
> Apologies for lack of clarity.
> - To tackle the uncertainty caused by the multi-modal characteristics of future trajectories, our framework performs prediction and planning while explicitly considering the multi-modal nature of the future. As described in Line 143 of the main paper, the predicted future BEV feature is multi-modal and shares the same modality as the ego feature used for planning. Accordingly, we also predict a multi-modal future BEV representation. For supervision, we adopt a winner-takes-all strategy: the best trajectory and the corresponding future BEV are selected using the same probability generated from the current ego feature, and the loss is computed based on the selected mode.
>
> - To address the uncertainty that may lead to errors in both prediction and planning, we propose the Iterative Scene Modeling and Vehicle Planning module. This design allows for iterative refinement of both future BEV prediction and trajectory planning, resulting in more accurate and consistent outcomes.
>
> - We will revise the paper to include a clearer and more detailed description of this design.
>
> ### **Limitation 3 and Question 3: The choice of loss weights.**
>
> - **Explanation**: Thanks, and we will clarify. The loss balancing among multiple objectives is not specially hand-tuned. Instead, we follow the characteristics of the NAVSIM dataset and prior works [1][2], by scaling the current BEV loss by a factor of 10 and reducing the future BEV loss weight to 0.1. For nuScenes, we use uniform loss weights, setting all to 1.
>
> - **Experiment**:  As suggested, we have now conducted a sensitivity analysis, as shown in the table below. The results indicate that SeerDrive is not particularly sensitive to the choice of loss weights ($\lambda_1$, $\lambda_2$, $\lambda_3$).
>
> | Loss weight | NC ↑ | DAC ↑ | TTC ↑ | EP ↑ | PDMS ↑ |
> | :--- | :---: | :---: | :---: | :---: | :---: |
> | $\lambda_1$=1, $\lambda_2$=1, $\lambda_3$=1 | 98.2 | 96.9 | 94.5 | 82.6 | 88.5 |
> | $\lambda_1$=5, $\lambda_2$=0.5, $\lambda_3$=1 | 98.4 | 96.9 | 94.8 | 83.1 | 88.8 |
> | $\lambda_1$=10, $\lambda_2$=0.1, $\lambda_3$=1 (Origin) | 98.4 | 97.0 | 94.9 | 83.2 | 88.9 |
>
> ### **Limitation 4: Gains over prior work.**
>
> - **Explanation**: We do not directly build upon previous methods such as DiffusionDrive [1] or WoTE [2], which are orthogonal to our approach. Nevertheless, our framework achieves state-of-the-art performance. Specifically, unlike DiffusionDrive [1], we do not adopt a diffusion-based policy for planning, and unlike WoTE [2], we do not employ multiple future BEV predictions for evaluation. We believe that incorporating such designs into our framework could potentially lead to further performance improvements.
>
> - **Gains compared to the baseline**: As demonstrated in Table 3 of the main paper, our proposed design improves the PDMS from 87.1 to 88.9, with a notable gain of +1.8. Additionally, as shown in Table 2, we conduct ablation studies on the nuScenes dataset, which clearly validate the effectiveness of our proposed modules through consistent performance improvements.
>
> - **Complexity**: Despite the iterative inference and dual-task training, our model does not introduce additional complexity. The average inference time is 24 ms and the parameter size is 66M, which is comparable to DiffusionDrive [1] and WoTE [2].
>
> ### **Question 1: Quantitative metric for predicted future BEV.**
>
> Great point! The mIoU of the predicted future BEV map on the NAVSIM dataset is 38.87. We will clarify.
>
> >[1] DiffusionDrive: Truncated Diffusion Model for End-to-End Autonomous Driving, CVPR 2025.
>
> >[2] End-to-end driving with online trajectory evaluation via bev world model, ICCV 2025.

---

### Official Review · Reviewer_W3tz · 2025-07-04

**Clarity:** 3
**Significance:** 3
**Originality:** 3
**Rating:** 5
**Confidence:** 4

**Summary:**

This paper addresses the limitations of existing end-to-end autonomous driving frameworks that typically operate in a one-shot manner, relying primarily on the current scene context without sufficiently considering the temporal evolution of the environment.

To address these issues, the authors introduce SeerDrive, a novel end-to-end framework designed to jointly model future scene evolution and trajectory planning in a closed-loop manner. The method consists of two principal components:
(1) Future-aware planning, which incorporates predicted future bird’s-eye view (BEV) scene features directly into the trajectory planner.
(2) Iterative scene modeling and vehicle planning, which jointly refines scene evolution predictions and planned trajectories through collaborative optimization.

Extensive experiments on the NAVSIM and nuScenes benchmarks demonstrate that SeerDrive achieves notably better performance than current state-of-the-art methods.

**Questions:**

1. See weaknesses.

2. What are the parameter size and inference cost of the proposed method?

**Ethical Concerns:**

["NO or VERY MINOR ethics concerns only"]

**Final Justification:**

The response effectively addresses my concerns about the originality compared to previous methods and the effectiveness of the proposed modules. The proposed approach to exploring the interaction between world modeling and planning seems well-justified, and the experiments on three popular benchmarks demonstrate its effectiveness.

**Limitations:**

Yes.

**Quality:**

3

**Strengths And Weaknesses:**

Strengths:

1. The proposed SeerDrive framework shows improved performance and achieves state-of-the-art results in mainstream benchmarks.
Ablation studies demonstrate the effectiveness of the proposed modules.

2. The paper is well-written, with clear notations, equations and figures, making it easy for readers to follow.

Weaknesses:

1. There are several existing methods leveraging scene future prediction as a latent world model to enhance planning, for example LAW [1], which leverages current sensor data and ego waypoints to predict future latent representation, then utilizes it as a prior for planning.
What are the significant contributions of the proposed framework, comparing with the existing works?

2. According to ablation studies, compared with the baseline, the proposed modules contribute limited improvement.

3. The paper provides failure cases and limitation analysis, while the discussion seems to be shallow. It is beneficial to provide more case studies and an in-depth discussion to guide future work.

[1] Enhancing end-to-end autonomous driving with latent world model. Li et al. ICLR 2025.

---

> ### Author Rebuttal · Authors · 2025-07-30
>
> We thank the reviewer for the detailed review and the suggestions for improvement. Below are our responses to the reviewer’s comments:
>
> ### **Weakness 1: Comparison with existing methods.**
>
> - Thanks for this comment. As discussed in Line 104 of the main paper, LAW [3] leverages world modeling as auxiliary supervision during training. In contrast, our method establishes feature-level interaction between world modeling and end-to-end planning, uniquely enabling iterative and closed-loop communication between the two processes. This design facilitates deeper integration, leading to significant improvements in planning performance.
>
> - We summarize in the table below to highlight the differences between our and prior works. We will further clarify in the revised paper.
>
> | Method            | Usage of predicted future scene from world model                                          |
> |-------------------|--------------------------------------------------------------------------------------------|
> | LAW [3]           | Used as an auxiliary supervision signal during training                                   |
> | SSR [4]           | Used as an auxiliary supervision signal during training                                   |
> | WoTE [2]          | Used to assist in selecting the best trajectory among multiple candidates                 |
> | SeerDrive (Ours)  | Used as feature-level reference for planning, enabling iterative interaction with planner |
>
> ### **Weakness 2: About ablation studies.**
>
> - We respectively disagree. As shown in Table 3 of the main paper, we conduct ablation studies on NAVSIM to evaluate the effectiveness of the proposed modules. By introducing Future-Aware Planning and Iterative Scene Modeling and Vehicle Planning, the PDMS (the most important metric) improves from 87.1 to 88.9, with a notable gain of +1.8, establishing a new state of the art (see Table 1 in main paper). Without these modules, the performance (87.1) is inferior to that of DiffusionDrive [1] (88.1), WoTE [2] (88.3), and Hydra-NeXt [5] (88.6).
>
> - As shown in Table 2 of the main paper, we also conduct ablation studies on nuScenes, which demonstrate clear performance gains from our proposed modules. In addition, we have now included another ablation experiment on nuScenes by removing all the proposed modules. The results below show a consistent and significant performance drop, further validating the effectiveness of our design.
>
> | Method | L2↓ 1s | 2s | 3s | Avg. | Col.↓ 1s | 2s | 3s | Avg. |
> |--------|--------:|----:|----:|------:|----------:|----:|----:|------:|
> | Baseline    | 0.40   |0.65| 1.02| 0.69  | 0.06    | 0.14| 0.37| 0.19 |
> | SeerDrive (Ours)    | **0.20**   | **0.39**| **0.69**|**0.43**  | **0.00**  |**0.05**|**0.14**| **0.06**  |
>
> ### **Weakness 3: Case studies and discussions.**
>
> - Thanks for this suggestion. In addition to the discussion provided in Line 262 of main paper, we will further add an in-depth discussion as follows:
>
> _The BEV world model adopts a transformer architecture specifically tailored to our framework, making it both effective and efficient for planning. However, it does not benefit from the generalization capabilities of foundation models. On the other hand, using off-the-shelf foundation models as world models often suffers from slow inference speed and challenges in joint optimization with the planner. Therefore, developing a tightly integrated paradigm of planning and world modeling represents a promising direction for future work._
>
> - We will add in-depth limitation analysis to guide future work in revision.
>
> - Regarding failure cases, beyond those presented in Line 20 of the appendix, we will include additional examples along with more detailed analysis in the revised version of the paper.
>
> ### **Question 2: Parameter size and inference cost.**
>
> Thanks, and we will provide it in the revision as follows:
>
> _The model size is 66M, and the average inference time is 24 ms on the NAVSIM dataset, under the same configuration as in the main paper._
>
> >[1] DiffusionDrive: Truncated Diffusion Model for End-to-End Autonomous Driving, CVPR 2025.
>
> >[2]  End-to-end driving with online trajectory evaluation via bev world model, ICCV 2025.
>
> >[3] Enhancing end-to-end autonomous driving with latent world model, ICLR 2025.
>
> >[4] Navigation-guided sparse scene representation for end-to-end autonomous driving, ICLR 2025.
>
> >[5] Hydra-NeXt: Robust Closed-Loop Driving with Open-Loop Training, arXiv 2503.12030

---

> > ### Comment · Reviewer_W3tz · 2025-08-05
> >
> > Thanks for your response. The proposed method to explore the interaction between world modeling and planning seems well-justified, and the experiments on three popular benchmarks demonstrate its effectiveness. I will raise my rating.

---

### Note · Authors · 2025-08-14

We sincerely thank the reviewers and AC for their time and valuable suggestions. Following the rebuttal and discussion, we summarize how we addressed the main concerns to facilitate the remaining review process.

The main concerns of **reviewer W3tz** are the comparison with existing methods, limited improvement in ablation, need for deeper discussion, and the parameter size and inference cost. We addressed these by listing similar methods and highlighting differences (**R1**), explaining improvements with an additional nuScenes experiment (**R2**), providing a deeper discussion of limitations and future work (**R3**), and clarifying model size and inference time (**R4**).

The main concerns of **reviewer LXsM** involve the single future prediction, lack of uncertainty modeling, hand-tuned loss balancing, limited gains, and the quantitative metric for predicted future BEV. We addressed these by explaining the rationale for single prediction with supporting ablation (**R1**), clarifying our uncertainty modeling (**R2**), justifying the loss design with sensitivity analysis (**R3**), analyzing gains over prior work (**R4**), and reporting the mIoU for predicted future BEV (**R5**).

The main concerns of **reviewer TxmL** are the comparison with WoTE and some structural and design issues. To address these, we first elaborated on the differences and connections between our method and WoTE (**R1**). Then, we clarified the structural and design issues, and provided additional ablation experiments to address the reviewer's concerns (**R2**, **R3** & **R4**).

The main concerns of **reviewer bKKZ** involve evaluating and visualizing future BEV prediction, testing in complex scenes, assessing the impact of iterations, and measuring inference time. We addressed these by reporting the mIoU of predicted future BEV and pointing to visualizations (**R1**), conducting experiments on longer horizons and complex urban settings in NAVSIM and Bench2Drive (**R2**), explaining the effect of iterations on NAVSIM with further ablation on Bench2Drive (**R3**), and clarifying inference time and real-time suitability (**R4**).

We acknowledge that our responses have been recognized as effectively resolving the reviewers’ concerns. These improvements have clearly strengthened the quality and clarity of our manuscript.

We hope this summary aids the remaining discussion and decision-making.

Thanks again for the review efforts.

Best regards,

The Authors

---

### Decision · Program_Chairs · 2025-09-17

**Decision:**

Accept (poster)

**Comment:**

The pre-rebuttal recommendation for this paper were rather divergent: 2 x Borderline Reject (`W3tz`, `bKKZ`), and 2 x Borderline Accept (`LXsM`, `TxmL`)

The main concerns expressed by the reviewers were the following:
- limited performance improvements
- novelty w.r.t. prior methods, e.g., WOTE
- some limited studies on quality of BEV maps, prediction of intermediate BEV maps, tuning of the loss weights, number of iterations
- unclear generalization to longer horizons and complex urban settings,

The authors have submitted detailed responses to address some of the concerns expressed by the authors in particular regarding novelty, computational complexity and implementation details. In addition, new closed-loop results on Bench2Drive with longer and more complex scenarii, show performance improvements over different baselines.
After the rebuttal reviewer `LXsM` has increased their score to Accept and reviewer  `bKKZ` to Borderline Accept.

Overall the reviewers appreciate the novelty of the contribution, bringing a fresh perspective in the area, the clean implementation, the consistent performance improvements and the ablation studies provided.

The meta-reviewer agrees with the assessment made by the reviewers and recommend this submission for acceptance.
This work has several merits, however the reviewers have pointed out a few directions of improvement. We encourage the authors to take into consideration the useful advice from the reviewers.